# Dual-Path Adversarial Generation Network for Super-Resolution Reconstruction of Remote Sensing Images

Zhipeng Ren [1,2], Jianping Zhao [1,*], Chunyi Chen [1], Yan Lou [3] and Xiaocong Ma [4]

1 School of Computer Science and Technology, Changchun University of Science and Technology, Changchun 130022, China
2 State Key Laboratory of Applied Optics, Changchun Institute of Optics, Fine Mechanics and Physics, Chinese Academy of Sciences, Changchun 130033, China
3 Institute of Space Optoelectronic Technology, Changchun University of Science and Technology, Changchun 130022, China
4 Muti-Media Communication Research Laboratory, University of Ottawa, Ottawa, ON K1N 6N5, Canada
* Correspondence: zjp@cust.edu.cn

**Abstract:** Satellite remote sensing images contain adequate ground object information, making them distinguishable from natural images. Due to the constraint hardware capability of the satellite remote sensing imaging system, coupled with the surrounding complex electromagnetic noise, harsh natural environment, and other factors, the quality of the acquired image may not be ideal for follow-up research to make suitable judgment. In order to obtain clearer images, we propose a dual-path adversarial generation network model algorithm that particularly improves the accuracy of the satellite remote sensing image super-resolution. This network involves a dual-path convolution operation in a generator structure, a feature mapping attention mechanism that first extracts important feature information from a low-resolution image, and an enhanced deep convolutional network to extract the deep feature information of the image. The deep feature information and the important feature information are then fused in the reconstruction layer. Furthermore, we also improve the algorithm structure of the loss function and discriminator to achieve a relatively optimal balance between the output image and the discriminator, so as to restore the super-resolution image closer to human perception. Our algorithm was validated on the public UCAS-AOD datasets, and the obtained results showed significantly improved performance compared to other methods, thus exhibiting a real advantage in supporting various image-related field applications such as navigation monitoring.

**Keywords:** remote sensing image; super-resolution; attention mechanism; navigation monitoring

## 1. Introduction

Satellite remote sensing imaging is a popular and efficient method to achieve indirect acquisition of surface information, by analyzing the radiation characteristics and electromagnetic wave reflection characteristics of ground objects. An ideal image is set to contain objects that are objectively and accurately identifiable, such as mountains, rivers, vehicles, and buildings. However, due to the complexity of the satellite imaging system circuit and unknown electromagnetic environment, the data acquisition process is often interfered by a bad natural environment, noise, and other factors, resulting in a lower quality of the output image. This deterioration of the quality can decrease the identification accuracy of the object and the usability of the image, thus introducing more difficulties to the follow-up research. Therefore, in current applications, additional preprocessing techniques such as noise reduction and super-resolution are usually required to improve image quality. This preprocessing is a critical stage as the data it generates serve as an information foundation can be relied on for any later earth observation, analysis, identification, and accurate judgment. For this reason, image super-resolution reconstruction technology has important research value and broad application prospects in the field of satellite remote sensing [1].

Deep learning is a branch of machine learning, which has a remarkable effect on speech and image recognition; it enables the machine to imitate human audiovisual perception and thinking, thereby solving many complex pattern recognition problems [2,3]. In particular, the image super-resolution reconstruction technology based on deep learning has made remarkable achievements in recent years. A super-resolution convolutional neural network (SRCNN) was proposed by Dong et al. [4] in 2014. This method extracts image features through convolution operation, and then optimizes model mapping parameters by continuously comparing low-resolution images with high-resolution images. As a result, a precedent was created for deep learning in the image super-resolution field. Later, an accelerated super-resolution model of a neural network (fast SRCNN, FSRCNN) was proposed by Dong et al. [5], which innovatively improved sample images by utilizing a deconvolution operation in the last layer in the model, to quickly construct the super-resolution of images with different scales. In 2016, Kim et al. [6] proposed a very deep super-resolution convolutional network (VDSR), which introduced residual learning to ensure that a combination of 20 convolution layers could be utilized to extract image features over a long distance, and the reconstruction results were significantly impressive. Lim et al. [7] proposed an enhanced deep convolutional network (EDSR) to learn the differences in detail between high-resolution images and low-resolution images. The trained model was significant in size and optimized with fewer modules in place. Similarly, to improve the overall performance of the network, Lai et al. [8] designed a deep Laplacian pyramid network for fast and accurate super-resolution (LapSRN). The Laplacian pyramid structure was adopted to obtain the reconstructed super-resolution image by layer up-sampling and fusion of residual image features. Ledig et al. [9] also proposed a super-resolution algorithm based on a generative adversarial network (SRGAN). The image reconstructed by SRGAN and the corresponding high-resolution image were input into a discrimination network to determine a true or false result. Through this confrontation training, the super-resolution reconstruction capability of the model was iteratively improved.

However, the existing super-resolution network algorithms have been built with relatively insufficient research on satellite remote sensing imaging, where room for further improvement still exists, particularly in terms of developing a strengthened recovery effect and a lowered complexity. How to improve the quality of satellite remote sensing imaging and provide accurate data for satellite navigation [10], surface observation [11], mapping [12], and military reconnaissance [13] has become a recent research hotspot. In this paper, we propose a dual-path generative adversarial network model algorithm which can be applied directly to the super-resolution reconstruction of satellite remote sensing images for improved quality and performance. Specifically, our system includes a dual-path convolution operation in the generator structure, a feature mapping attention mechanism to extract important feature information from a low-resolution image, an enhanced deep convolutional network to extract the deep feature information of the image, and a construction layer to fuse the deep feature information and the important feature information together for the final output. The improved algorithm structure consisting of a loss function and discriminator also achieves a relatively optimal balance between the output image and the output of the discriminator, thus bringing the final result closer to human perception.

## 2. The Related Work

In this section, we describe the details of the SRGAN network architecture, a model that utilizes adversarial training thinking during the image super-resolution task to provide more abundant image details and improved image authenticity in the output. The model is divided into two sub-models, namely, G (generator) and D (discriminator). The overall framework of the SRGAN algorithm is shown in Figure 1, where the G network takes low-resolution images as input, and then generates targeted output that is close to ground-truth images. This basically describes the entire process of the image super-resolution reconstruction task. Next, the D network is used to assess the main task output for achieving

refined results by making judgement on a true or false basis. The reconstructed image of the G network is compared with the corresponding ground-truth image. Through this confrontation training, the reconstruction effect of the model is constantly improved.

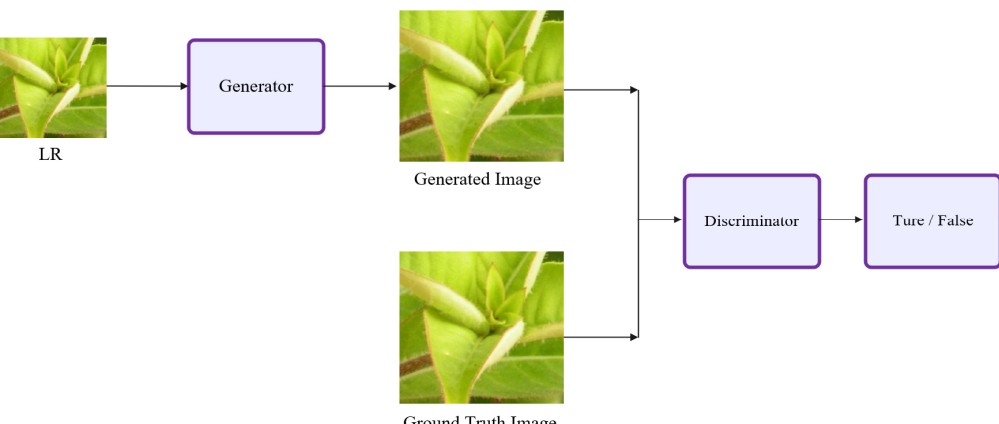

**Figure 1.** The frame diagram of SRGAN.

The SRGAN model is different from that of the traditional GAN. The GAN is random noise, whereas the SRGAN is a low-resolution image. The generator network structure is shown in Figure 2; the core structure of the network is a residual module composed of two $3 \times 3$ convolution layers, a standard regularization (BN) layer [14], and a ReLU activation function. At the end of the network, a sub-pixel convolution layer is used to up-sample the image to obtain a reconstructed image of identical size to the original image. The residual module structure is shown in Figure 3.

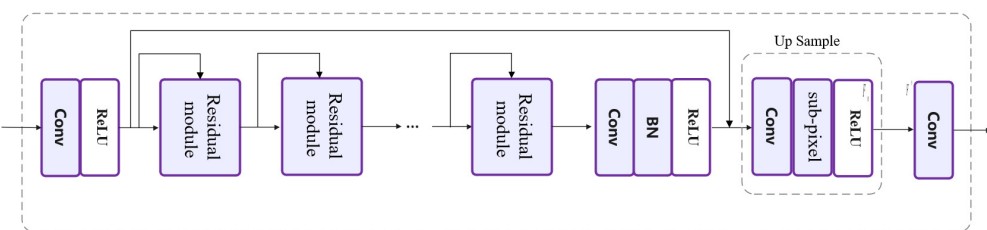

**Figure 2.** Generator network structure.

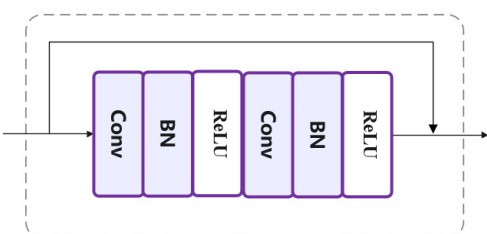

**Figure 3.** Residual module structure.

The core module of the discriminant network in the SRGAN model is different from the core of the generator network. The core module is composed of a convolution layer, a Leaky ReLU activation function, and a standard regularization BN layer. A diagram of the discriminator is shown in Figure 4; after the discriminator receives image information as its input, it passes it through its Sigmoid function, which a makes judgement and classifies the results into true or false categories.

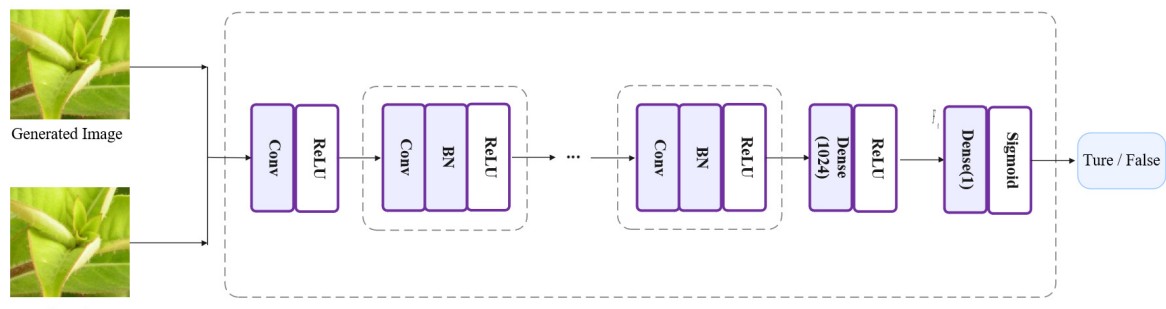

**Figure 4.** Discriminator structure.

In order to improve the final image generation effect in the SRGAN network, the composition of the loss function is changed, and the concept of perceptual loss [15] is introduced. In the original SRGAN, loss is defined as the sum of different weights of content loss and adversarial loss, as shown in Equation (1), where $L_x^{SR}$ is the content loss, $L_{Gen}^{SR}$ is the adversarial loss, and $L^{SR}$ is the final fusion loss.

$$L^{SR} = L_x^{SR} + 10^{-3} L_{Gen}^{SR}. \tag{1}$$

The definition of the pixel-by-pixel MSE loss function is shown in Equation (2).

$$L_{MSE}^{SR} = \frac{1}{r^2 HW} \sum_{x=1}^{rH} \sum_{y=1}^{rW} (I_{x,y}^{HR} - G_{\theta_G}(I^{LR})_{x,y})^2, \tag{2}$$

where $H$ and $W$ represent the size of the matrix, $r$ represents the up-sample scale, and $I^{HR}$ and $I^{LR}$ are the high-resolution and low-resolution images, respectively. The MSE loss function is often used as the convergence target in the field of super-resolution reconstruction; in particular, in the SRGAN model, a concept of loss close to human perception is proposed. More specifically, this concept is defined by the direct Euclidean distance between the $G_{\theta_G}(I^{LR})$ and $I^{HR}$ features of the super-resolution image extracted by VGG [16], and the loss function of VGG is defined as follows:

$$L_{VGG/i,j}^{SR} = \frac{1}{H_{i,j} W_{i,j}} \sum_{x=1}^{H_{i,j}} \sum_{y=1}^{W_{i,j}} (\phi_{i,j}(I^{HR})_{x,y} - \phi_{i,j}(G_{\theta_G}(I^{LR}))_{x,y})^2. \tag{3}$$

$H_{i,j}$ and $W_{i,j}$ represent the dimension values of the feature map in the feature extraction network. In the VGG feature extraction network, $\phi_{i,j}$ represents the characteristic diagram before the $i$th largest pooling layer and after the activation of the $j$th convolution layer.

The generation loss of the generator subnetwork is added to the perception loss item in the network, and its purpose is to cheat network D as much as possible. The D network constantly refines the ability to distinguish the output of the G network against the high-definition original image in training. The definition of adversarial loss $L_{Gen}^{SR}$ is shown in Equation (4).

$$L_{Gen}^{SR} = \sum_{n=1}^{N} -\log D_{\theta_D}(G_{\theta_G}(I^{LR})). \tag{4}$$

## 3. The Proposed Network Model

Remote sensing images are quite different from natural images in that they require more sufficient reconstruction of edge details, since they contain extensive ground object information, such as forests, trees, buildings, mountains, and rivers. The current challenge for GAN in the field of remote sensing image super-resolution is that some models based on pixel level loss will cause the reconstructed image to become too smooth and not close to the quality perceived by human eyes; however, with the support of the GAN network,

one can properly solve this problem by obtaining more abundant boundary information from its reconstructed image. In particular, our solution to combat this issue is a dual-path adversarial generation network model algorithm, the overall framework of which is shown in Figure 5. In this paper, we redesign the structure of the generator module, the parameter setting and activation function. We also revise and adjust the training parameters to obtain improved results from the loss function and the discriminator. The preliminary results show that the output image and discriminator are relatively optimized and balanced, and the super-resolution reconstruction of the remote sensing image is capable of achieving results as we expect.

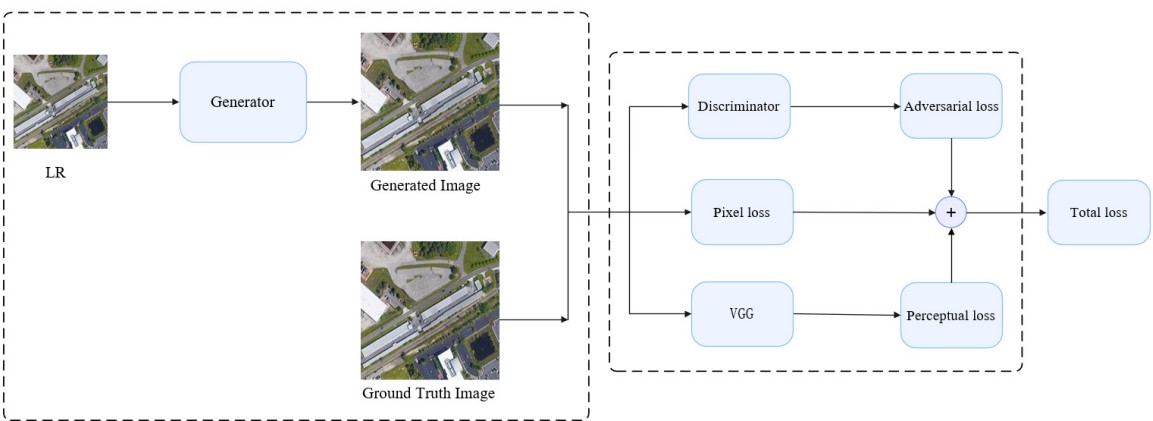

**Figure 5.** The overall framework of the proposed network.

### 3.1. Generator Structure

The generator structure designed in this paper is shown in Figure 6. Here, we adopt a dual convolution operation; a feature mapping attention mechanism is introduced to extract the important feature information of the low-resolution image and transfer it between channels on one path, whereas an EDSR deep residual convolution is introduced directly to extract the comprehensive feature information of the image on the other path. The image information extracted from the comprehensive feature is then fused together with the important feature information in the reconstruction layer to restore the image closer to the perceived quality of human eyes. The extracted important feature information, which serves as a refining element, makes up for the deficiency of deep residual network feature extraction.

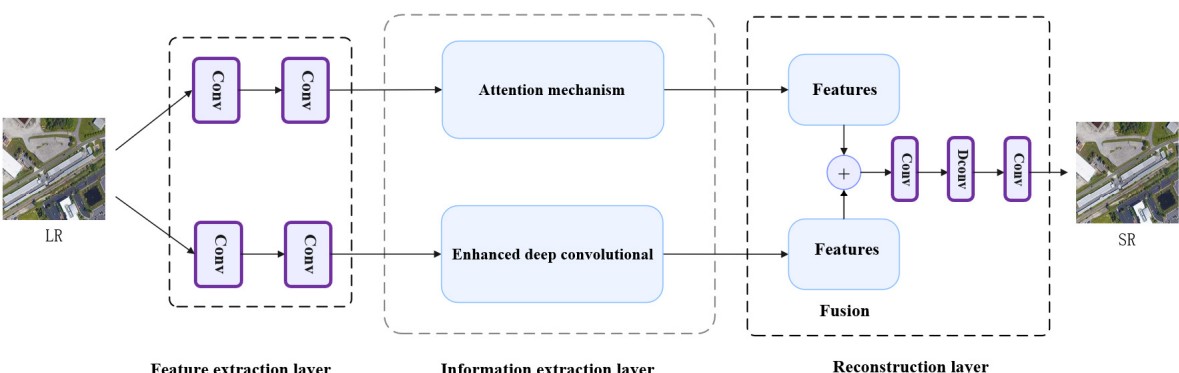

**Figure 6.** The architecture of the generator network.

The channel attention mechanism not only plays a significant role in improving the reconstruction accuracy of the network, but also has high module portability, which can be easily loaded into the existing network structure. It distinguishes the significance of image channels by giving different weights to image channels through learning. This

allows the network to pay more attention to the rich information channels and increases the feature extraction ability of the network that is also instructive for image reconstruction tasks. Therefore, this article utilizes the channel attention mechanism for image super-resolution reconstruction (SRCAN) [17,18] As shown in Figure 7, the network is set as five submodules, where each 3 × 3 convolution operation is followed by an attention mechanism submodule. In addition, the number of modules can be constantly adjusted according to the reconstruction effect.

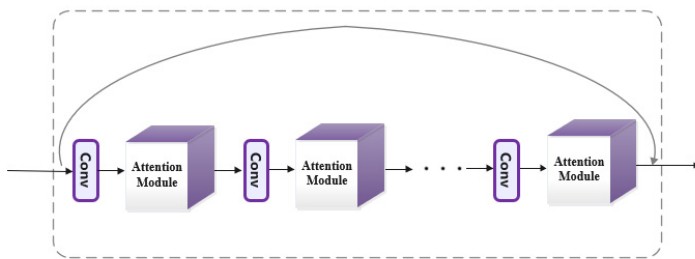

**Figure 7.** The architecture of the SRCAN module.

The enhanced deep convolutional network module includes five submodules. Each submodule contains three convolution operations and introduces local residuals. Before the information flow enters each submodule, we utilize a 1 × 1 convolutional operation to the output of the previous level to perform fusion compression. As shown in Figure 8, local residual learning and global residual learning are conducted within the network. In terms of the scope, local refers to residual learning within the submodule, and global refers to residual learning between the input and output of the entire network.

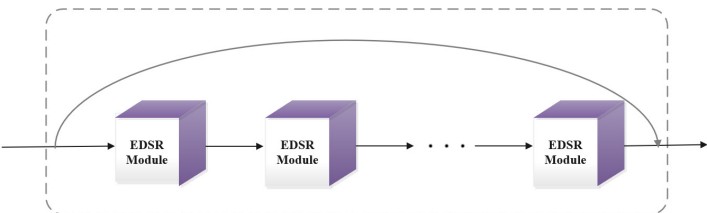

**Figure 8.** The architecture of the EDSR module.

In addition, the traditional activation function has a drawback that needs to be modified for achieving better results. Although the ReLU activation function has high calculation efficiency and can make the network converge quickly, the traditional ReLU function leads to the death of neurons. When the input is close to 0 or in a negative interval, the gradient disappears, and the network is unable to carry out the backpropagation of the gradient, resulting in the network being unable to carry out normal learning. This can be expressed as follows:

$$f(y) = \begin{cases} y & y \geq 0 \\ 0 & y < 0 \end{cases}. \tag{5}$$

In the situation where $y$ is less than 0, the ReLU activation function leads to the problem of neuron death. Through our research to prevent this from happening, we found that the Leaky ReLU activation function has a nonzero positive slope in the negative axis interval, which is capable of carrying out backpropagation by implementing the property of the ReLU function while preventing the above issue. This can be expressed as follows:

$$f(y) = \begin{cases} y & y \geq 0 \\ \delta y & y < 0 \end{cases}, \tag{6}$$

where $\delta$ is defined as a real number within an interval (0, 1), and the output of the function is ensured to preserve its sample information within this negative input region.

### 3.2. Discriminator Structure

The structure of the discriminator is shown in Figure 9. The discriminator is the same as that in the SRGAN. The discriminator contains eight convolutional blocks together as a structure, as shown in the figure. The infrastructure in each module is also shown. To avoid the problem of neuron death, Leaky ReLU is selected as the activation function of the network. The convolution core of each convolution layer in the network is set to $3 \times 3$, followed by a number of channels increasing from 64 to 512 and a doubled rate compared to the original. At the end of the network, two full connection layers and one sigmoid activation function are introduced to obtain the probability of sample data classification. Mathematically, a discriminator network is considered to composed of the following form of a residual network:

$$f(x, \theta) = F(x, \{W_i\}) + x, \tag{7}$$

where $x$ is the input value, $\theta = \{W_1, W_2, \ldots, W_i\}$ is the parameter set, and $F(x, \{W_i\})$ is the residual network to be trained. For example, if $F$ represents $W_i h(W_{i-1} x)$ and $h$ represents the activation function, then the final input of the discriminator ignoring bias can be represented as $D(x, \theta) = H(f(x, \theta))$, where $H$ refers to the distance metric difference.

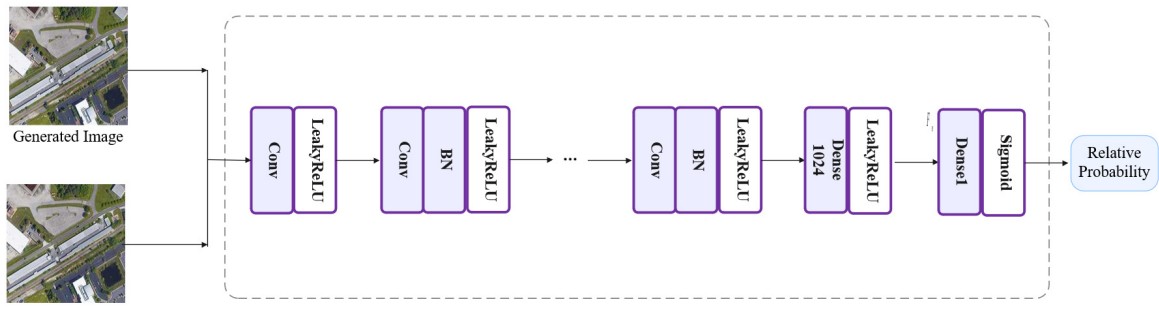

**Figure 9.** The architecture of the discriminator module.

The loss function exists in the form of fusion loss. In addition to the pixel-by-pixel loss defined by $L_1$ loss, it includes perceptual loss and adversarial loss. The final fusion loss is obtained by balancing and accumulating the weight coefficients of different losses. The pixel loss function is shown in Equation (8).

$$L_1 = \frac{1}{WHC} \sum_i^W \sum_j^H \sum_k^C \| I_{i,j,k}^h - I_{i,j,k}^g \|, \tag{8}$$

where $H$ and $W$ represent the size of the matrix, and $C$ represents the number of channels. In addition to pixel loss, the loss in this paper refers to the perception loss concept proposed in the SRGAN. As previously mentioned, SRGAN uses the pretrained VGG network to extract both the G network super-resolution reconstruction results and the high-definition image features. This can be defined as the Euclidean distance between the extracted features, as shown in Equation (9).

$$L_{\text{VGG}/i,j}^{SR} = \frac{1}{H_{i,j} W_{i,j}} \sum_{x=1}^{H_{i,j}} \sum_{y=1}^{W_{i,j}} \left( \phi_{i,j}(I^{HR})_{x,y} - \phi_{i,j}(G_{\theta_G}(I^{LR}))_{x,y} \right)^2, \tag{9}$$

where $G_{\theta_G}(I^{LR})$ represents the super-resolution reconstruction image of the generator subnetwork. As mentioned above, $H_{i,j}$ and $W_{i,j}$ represent the dimension values of the feature map in the feature extraction network, inside the VGG feature extraction network, while $\phi_{i,j}$ represents the characteristic diagram before the $i$th largest pooling layer and after the activation of the $j$th convolution layer.

The GAN adversarial loss $L^{SR}_{Gen}$ is defined by Equation (10).

$$L^{SR}_{Gen} = \sum_{n=1}^{N} - \log D_{\theta_D}(G_{\theta_G}(I^{LR})).$$

(10)

Therefore, the final total loss function is defined by Equation (11).

$$L^{SR} = \eta L^{SR}_{\text{VGG}/i,j} + \lambda L^{SR}_{Gen} + L_1,$$

(11)

where $\eta$ and $\lambda$ are the weight coefficients used for balancing before different loss functions, which were set to 3 and 0.25, respectively, in our experiment.

*3.3. Network Training*

The training process of the model is mainly a composition of processes including the initialization of the program parameters, the accessing of the training datasets, generator reconstruction and optimization, discriminator identification and optimization, and the storing of the training records and model parameters. The whole training process can be described by the following steps:

(1) Initialize program parameters, and access the training datasets.
(2) Input the low-resolution images in the generator network, and reconstruct the high-resolution images through the dual-path module.
(3) Calculate and save the PSNR, the SSIM, and the running time.
(4) Compare the output results with the corresponding ground-truth HR to calculate the loss function in the discriminator network.
(5) If the discriminator result is FALSE, then update the weight parameter by backpropagation, and return to step (2).
(6) If the discriminator result is FALSE, then judge whether the loss function is convergent. If it is not convergent, optimize the discriminator. If it is, return to step (2).
(7) Obtain the optimized generator and discriminator network model.

In this article, we selected 600 images from the UCAS-AOD remote sensing image database as the dataset. The UCAS-AOD dataset was released by the University of Science and Technology of China in 2014, and it contains background samples of cars, aircraft, buildings, mountains, and rivers, including 400 pictures that we chose as our training dataset. Firstly, we down-sampled each picture three times, and then expanded the training dataset through vertical flip, horizontal flip, and rotate operations. Next, we cut the images into $256 \times 256$ size fragments paired together with the ground-truth images, and we stored all the fragments in an overall training dataset after sorting them in order [19–21]. Next, we set the momentum parameter to 0.9 and the weight attenuation to $10^{-4}$ [22,23]. We also set the initial learning rate to 0.1, and then reduced it 10-fold for every 25 cycles [24–26]. Each group of 64 training data was packed as a batch and sent into the network, where its output was passed through the designed generator and then fused at the reconstruction layer. At this point, we chose the slope of the activation function as 0.2 [27–29]. The network then performed a comparison using the output results against the original graph in the discriminator, continuously optimized the parameters in the generator and discriminator, and saved each generation of training network to be utilized in the next refining iteration. In our experiment, the number of iterations was set to 100, and the total training cost was about 165 h. Upon completion, we were able to select the trained model for testing according to the convergence effect of the loss function. Eventually, we selected the 90th generation network model for our subsequent testing experiments.

The ultimate goal of the network is to achieve a Nash equilibrium between the G subnetwork and the D subnetwork. In the training process, it is likely for one side to have a good training effect, while the other side cannot converge. To visualize the change process of the loss during the experiment, their curves are shown in Figure 10.

Our experiment evaluated the training results by observing the convergence curve of the loss during the training process. It can be seen from the loss curve figures that the GAN loss was initially in an unstable state; however, with the increase in the number of iterations, the model continued to fit the input data, while the perceptual loss and pixel loss gradually converged. As the subnetwork improved its ability to converge, it gradually reached a dynamic equilibrium state, and the fusion loss of the whole network also tended to converge in a similar way.

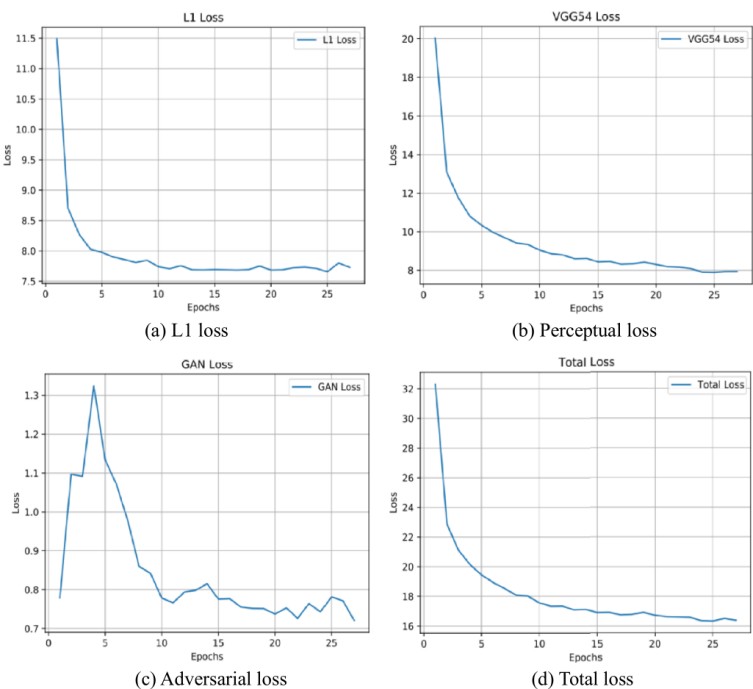

**Figure 10.** The training curves of the network.

## 4. Experiments and Comparisons

### 4.1. Experimental Hardware Configuration

We set up a 64-bit Linux operating system with a CUDA 8 toolkit, Cudnn 5.0, PyCharm, Pytorch, and Matlab v2016a as our experiment's software environment. The hardware environment of the computer included an NVIDIA GeForce GTX 1060 GPU, 16 GB RAM, an Intel Core i7-3770 CPU running at 3.4 GHz, and 1 TB of hard disk storage space.

### 4.2. Experimental Comparison

We divided the remain 200 graphs datasets into four test datasets, containing 20, 30, 50, and 100 pairs of images, and we labeled them as Dataset 1, Dataset 2, Dataset 3, and Dataset 4, respectively. The trained model in this paper was sequentially tested using the images of each dataset, and our testing results were evaluated using the following objective reconstruction and subjective visual evaluation indicators: the peak signal-to-noise ratio (PSNR) [30], the structural similarity (SSIM) [31], and the mean opinion score (MOS) [9]. After finishing running the tests, we obtained a series of results. The average PSNR indicators of Dataset 1, Dataset 2, Dataset 3, and Dataset 4 were 31.15 dB, 31.72 dB, 32.06 dB, and 32.64 dB, and the average SSIM indicators were 0.8034, 0.8142, 0.8179, and 0.8255, respectively. In order to prove that our model algorithm was superior compared to other algorithms, we also conducted simulation experiments using SRCNN [4], FSRCNN [5], VDSR [6], EDSR [7], LapSRN [8], and SRGAN [9]. As shown in Table 1, the structures of the related networks are displayed in terms of the training time and the number of parameters of each model.

**Table 1.** Structural analysis of each super-resolution algorithm.

| Algorithm | Input | Reconstruction | Depth | Filters | Parameters | Residual Structure | Loss | Training Time |
|---|---|---|---|---|---|---|---|---|
| SRCNN | LR + bicubic | Direct | 3 | 64 | 57 K | No | L2 | 75 h |
| FSRCNN | LR | Direct | 8 | 56 | 12 K | No | L2 | 46 h |
| VDSR | LR + bicubic | Direct | 20 | 64 | 665 K | Yes | L2 | 4 h |
| MDSR | LR | Direct | 162 | 64 | 8000 K | Yes | Charbonnier | 160 h |
| LapSRN | LR | Progressive | 24 | 64 | 812 K | Yes | Charbonnier | 72 h |

The objective evaluation index result for each network model was recorded as shown in Table 2. Impressively, the PSNR value generated by our algorithm was superior to that of other advanced algorithms. On Dataset 2 with three amplifications, the average PSNR index of our algorithm was 0.46 dB higher than SRGAN's result and 0.96 dB higher than EDSR's result. Similarly, it is also clear on Dataset 4 that the average PSNR index generated by our algorithm was 0.82 dB higher than VDSR's value and 2.73 dB higher than SRCNN's value. In Table 3, the average SSIM result calculated using our algorithm on Dataset 3 with three amplifications was 0.005 higher than EDSR's result and 0.0332 higher than FSRCNN's result. The average SSIM result calculated using our algorithm was 0.009 higher than VDSR's value and 0.0396 higher than SRCNN's value on Dataset 4. Figure 11 shows the values of PSNR and SSIM on Dataset1 and Dataset4 with different methods, which directly indicates that our model has advantages over others.

**Table 2.** Average PSNR on Dataset 1, Dataset 2, Dataset 3, and Dataset 4 with different algorithms.

| Dataset | Scale | Bicubic | SRCNN | FSRCNN | VDSR | EDSR | LapSRN | SRGAN | Ours |
|---|---|---|---|---|---|---|---|---|---|
| **Dataset 1** | **3×** | 26.54 | 28.43 | 28.56 | 30.37 | 30.23 | 30.34 | 30.70 | **31.15** |
| **Dataset 2** | **3×** | 26.59 | 28.98 | 29.10 | 30.91 | 30.76 | 30.87 | 31.26 | **31.72** |
| **Dataset 3** | **3×** | 27.36 | 29.52 | 29.66 | 31.47 | 31.32 | 31.43 | 31.61 | **32.06** |
| **Dataset 4** | **3×** | 27.83 | 29.91 | 30.04 | 31.82 | 31.65 | 31.77 | 32.18 | **32.64** |

**Table 3.** Average SSIM on Dataset1, Dataset2, Dataset3 and Dataset4 with different algorithms.

| Dataset | Scale | Bicubic | SRCNN | FSRCNN | VDSR | EDSR | LapSRN | SRGAN | Ours |
|---|---|---|---|---|---|---|---|---|---|
| **Dataset 1** | **3×** | 0.7543 | 0.7781 | 0.7792 | 0.7912 | 0.7915 | 0.7913 | 0.8002 | **0.8034** |
| **Dataset 2** | **3×** | 0.7549 | 0.7795 | 0.7813 | 0.7959 | 0.7957 | 0.7953 | 0.8114 | **0.8142** |
| **Dataset 3** | **3×** | 0.7615 | 0.7836 | 0.7847 | 0.8134 | 0.8129 | 0.8126 | 0.8137 | **0.8179** |
| **Dataset 4** | **3×** | 0.7687 | 0.7859 | 0.7866 | 0.8165 | 0.8158 | 0.8157 | 0.8189 | **0.8255** |

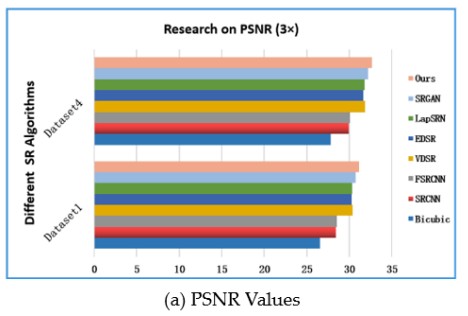

(a) PSNR Values

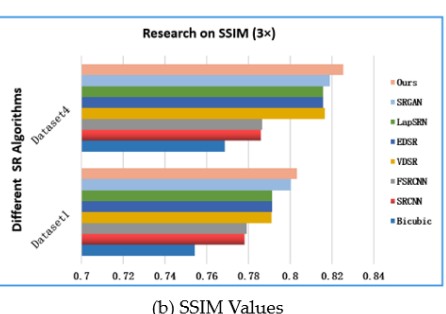

(b) SSIM Values

**Figure 11.** Compared PSNR and SSIM results with a 3× scale on Dataset 1and Dataset 4.

For the subjective intention score MOS, we invited 50 students from the laboratory to conduct a subjective survey by giving their scores, and then calculated the average values. The score reference was based on the following grades: excellent (5 points), good (4 points), general (3 points), poor (2 points), and extremely poor (1 point). As shown in Table 4 and Figure 12, we can clearly identify that our algorithm not only had a significant recovery

effect on observable evaluation indicators, but also had advantages in subjective evaluation according to the subjective intention scores rated by students on each algorithm.

**Table 4.** Average MOS values with different algorithms on Dataset 1, Dataset 2, Dataset 3, and Dataset 4.

| Dataset | Scale | Bicubic | SRCNN | FSRCNN | VDSR | EDSR | LapSRN | SRGAN | Ours |
|---------|-------|---------|-------|--------|------|------|--------|-------|------|
| Dataset1 | 3× | 1.88 | 2.45 | 2.63 | 3.12 | 3.19 | 3.28 | 3.34 | **3.41** |
| Dataset2 | 3× | 1.76 | 2.31 | 2.55 | 3.06 | 3.15 | 3.17 | 3.26 | **3.35** |
| Dataset3 | 3× | 1.53 | 2.24 | 2.48 | 2.97 | 3.01 | 3.06 | 3.11 | **3.17** |
| Dataset4 | 3× | 1.63 | 2.28 | 2.37 | 2.99 | 3.07 | 3.08 | 3.19 | **3.23** |

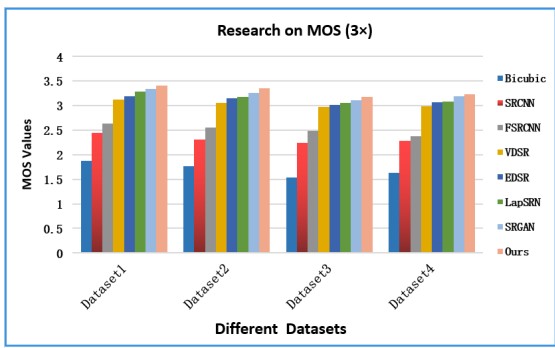

**Figure 12.** Compared MOS results with a 3× scale on Dataset 1, Dataset 2, Dataset 3, and Dataset 4.

### 4.3. Experimental Analysis

In this section, we analyze our designed algorithm from the perspective of subjective vision. As shown in Figures 13–16, we again conducted the sensory comparison with the six state-of-the-art algorithms mentioned previously in this paper. Our method was capable of effectively restoring the lines, edges, and details of the image, thus improving the resolution ratio of the remote sensing images. The experimental values of the 3× "P0283" airport picture in Dataset 1 is shown in Figure 13. The result achieved using our algorithm showed a better visual experience on the reconstructed aircraft wings with a clearly defined background. The edges and lines of the wing were relatively uniform and smooth. Similarly, the shadows on the ground were also clearly visible. It is not difficult to see that, when our algorithm was applied to the reconstruction of the image, particularly when restoring edges on objects and differentiating them from their backgrounds, the lines between the grass and the road in the middle of the image, as well as the white lines on the right side of the image, all appeared straight, clear, and hierarchical, while the results obtained using the other algorithms still had various degrees of blur or distortion.

Figure 14 shows the experimental values calculated from the 3× "N0566" urban picture using Dataset 2. The image recovered using our method was closer to the ground-truth image. The outline of the small light rail train was clearly visible, and the oil tanker beside it was also relatively easy to identify. The lines and corners of tall buildings in the figure were displayed intuitively. On the other side, the reconstruction effect of SRCNN was vague, and objects could hardly be recognized. The LapSRN and SRGAN algorithms also produced results with limited clarity, particularly in the contours and lines of small trains.

Figure 15 shows the values of the experiment on the 3× "N0502" house picture using Dataset 3. The images restored by VDSR and EDSR showed a little distortion along the line of the roof. SRCNN, FSRCNN, and other algorithms also had shortcomings when restoring car windows, edges, and bodies. In contrast, lines between the side windows of the car on the far right could be clearly identified from the image obtained using our method, and the outline of the roof was more intuitive.

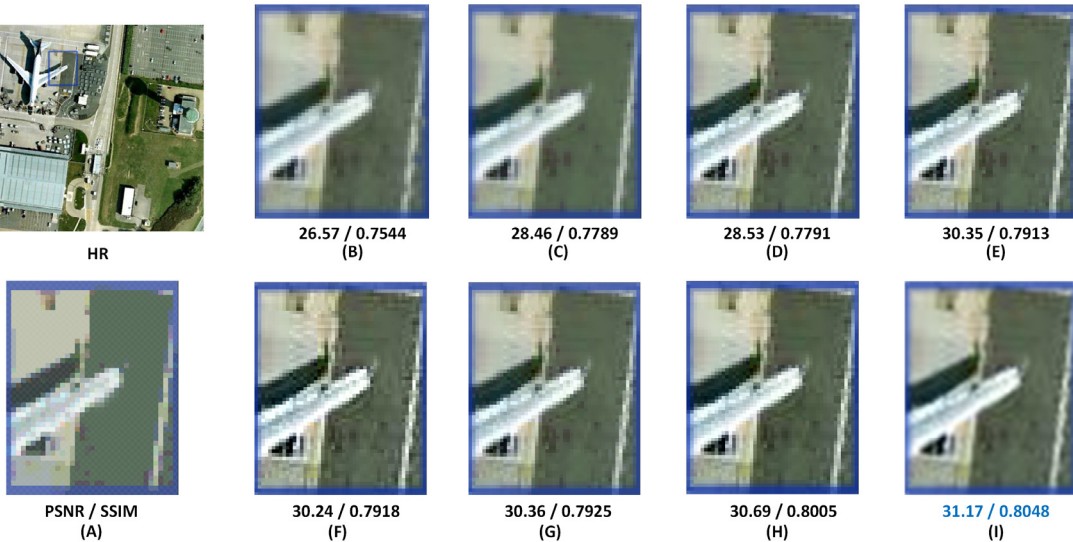

**Figure 13.** Comparison of values with different algorithms on the 3× "P0283" airport picture: (**A**) LR; (**B**) bicubic; (**C**) SRCNN; (**D**) FSRCNN; (**E**) VDSR; (**F**) EDSR; (**G**) LapSRN; (**H**) SRGAN; (**I**) ours.

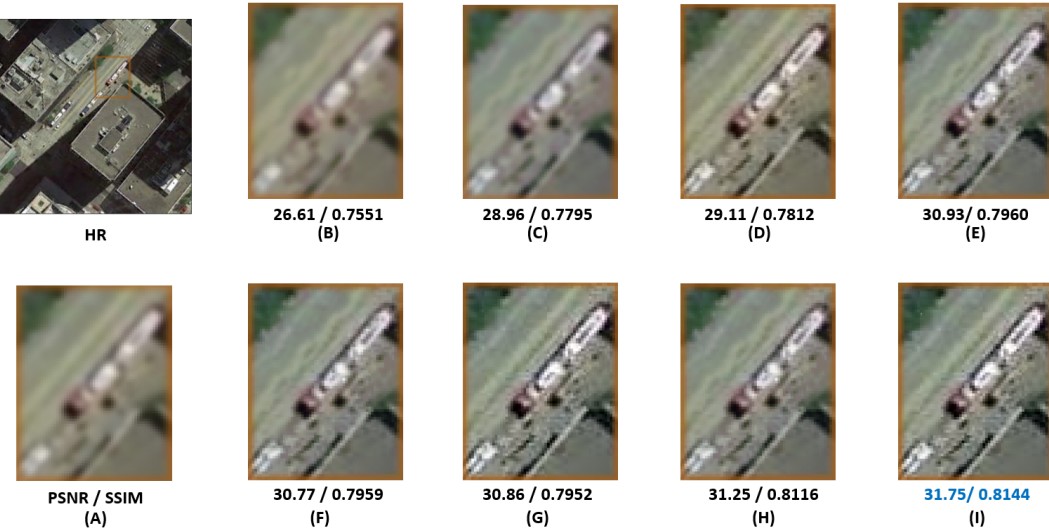

**Figure 14.** Comparison of values with different algorithms on the 3× "N0566" urban picture: (**A**) LR; (**B**) bicubic; (**C**) SRCNN; (**D**) FSRCNN; (**E**) VDSR; (**F**) EDSR; (**G**) LapSRN; (**H**) SRGAN; (**I**) ours.

Figure 16 shows the values of the experiment calculated from the 3× "P0808" airport picture using Dataset 4. The image reconstructed using the FSRCNN and EDSR algorithms led to a distortion of the aircraft with added deformation and artefacts, and these algorithms could not completely restore the shadow part of the aircraft tail to their normal shapes. The reconstructed image obtained using our method, on the other hand, could clearly show not only the white lines on the ground between the two aircrafts, but also the tail shadow of the aircraft. Therefore, our proposed algorithm in this paper proves its clear advantage in restoring lines, edges, and details to a degree closer to the real image.

Below, we introduce the limitations of our research. Because we need to spend more time on training and experiments for each algorithm, we only trained them with a 3× magnification of remote sensing images on a single scale. The model can be retrained again if the training datasets are changed, which will inevitably lead to changes in the simulation values and bring certain uncertainties to subsequent studies. In the future, we will work on how to make the algorithm adapt to different sizes of images and be better compatible with different training datasets.

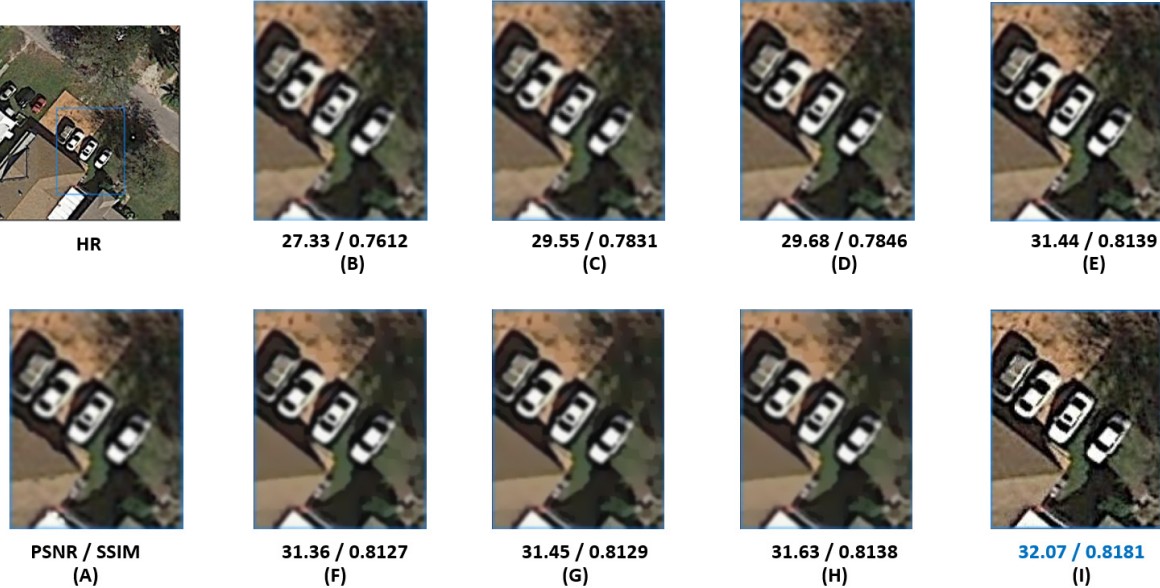

**Figure 15.** Comparison of values with different algorithms on the 3× "N0502" house picture: (**A**) LR; (**B**) bicubic; (**C**) SRCNN; (**D**) FSRCNN; (**E**) VDSR; (**F**) EDSR; (**G**) LapSRN; (**H**) SRGAN; (**I**) ours.

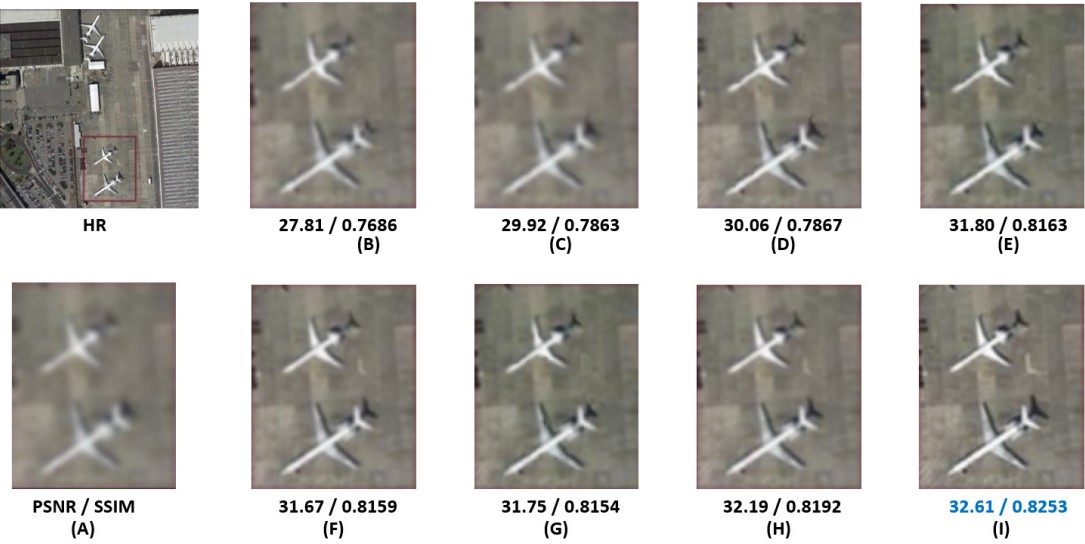

**Figure 16.** Comparison of values with different algorithms on the 3× "P0808" airport picture: (**A**) LR; (**B**) bicubic; (**C**) SRCNN; (**D**) FSRCNN; (**E**) VDSR; (**F**) EDSR; (**G**) LapSRN; (**H**) SRGAN; (**I**) ours.

## 5. Conclusions

In this paper, a new algorithm using a dual-path adversarial generation network model is proposed for the reconstruction of satellite remote sensing images. In the generator module, we utilized both the attention mechanism network and the enhanced deep convolutional network to extract the features from images, and we fused them in the reconstruction layer to generate high-resolution images. Moreover, the activation function and loss function were improved, such that the output image and discriminator could achieve a relatively optimal balance and restore a clear image. Additionally, we tested our algorithm on the public UCAS-AOD datasets, and then compared it with several relevant algorithms; our experimental results show that the algorithm in this paper could achieve a clear reconstruction of lines, edges, and details from satellite remote sensing images, thus greatly enhancing the quality of satellite remote sensing images. The average PSNR value was 32.64 dB on Dataset 4, which was 2.73 dB higher than on the SRCNN algorithm. The

improvement in data accuracy provided by our solution brings enormous performance potential for many practical applications such as satellite navigation, surface observation, mapping, and military reconnaissance. For the next step of our plan, deep learning technology can be applied to the field of satellite infrared remote sensing monitoring, to provide strong technical support for night military reconnaissance and intelligent clear imaging of space stations.

**Author Contributions:** Z.R., conceptualization, methodology, and writing; J.Z., supervision and review; C.C., validation; Y.L., training and testing; X.M., testing and review. All authors have read and agreed to the published version of the manuscript.

**Funding:** This research was funded by [the Department of Science and Technology of Jilin Province] grant number [20220402014GH], [the Education Department of Jilin Province] grant number [JJKH20220774KJ], [the Open Fund Project of the State Key Laboratory of Applied Optics] grant number [SKLAO2022001A07], [the Education and Science Planning Project of Jilin Province] grant number [GH22197], [the Natural Science Foundation of Jilin Province] grant number [YDZJ202201ZYTS411].

**Informed Consent Statement:** Informed consent was obtained from all subjects involved in the study.

**Data Availability Statement:** The dataset can be downloaded at the website: https://opendatalab.com/102/download (accessed on 16 July 2022).

**Acknowledgments:** This research was funded by the 2022 International Cooperation Project in Jilin Province Science and Technology Department (grant number 20220402014GH), the Science and Technology Research Project in Jilin Provincial Department of Education (grant number JJKH20220774KJ), the Open Fund Project of the State Key Laboratory of Applied Optics (grant number SKLAO2022001A07), and the Jilin Provincial Education Planning Project (grant number GH22197)].

**Conflicts of Interest:** The authors declare no conflict of interest.

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
