# Peer review of "Dual-Path Adversarial Generation Network for Super-Resolution Reconstruction of Remote Sensing Images"

_applsci, doi:10.3390/app13031245_

Round 1

Reviewer 1 Report

1- The English grammar should be further checked.

2- The review of the related super-resolution for remote sensing image processing is not comprehensive. The author should add some most popular papers in this field, for example, 10.1016/j.compeleceng.2017.02.012,https://doi.org/10.3390/rs14215423,10.1016/j.earscirev.2022.104110.

3- in the result section, please add the training time and the number of parameters of each of the models.

4- The most important is that I miss the discussion section; where the weaknesses and strengths of the proposed method are discussed.

Author Response

1.We revised the paper and replied all the points which provided from reviewers.
2.We improved the abstract and conclusion. 
3.We also checked and enhanced the level of English writing. We clarified the motivation of the proposed method and gave the details of motivations in the article.
4.We checked and enhanced the level of paper presentation. So that the description of manuscript is clear for potential reader and other researchers. 
5. We checked and updated all the references. 
6. We checked all parameters and equations, amended some related description of primary parameters. 
7. we rephrased the marked places in the paper for lower the similarity.

Author Response

(The authors gave the same response as above.)

Round 2

Reviewer 1 Report

The authors have addressed my concerns